# The Impact of Japan’s Soft Lockdown on Depressive Symptoms among Community-Dwelling Older Adults

**DOI:** 10.3390/healthcare11091239

**Published:** 2023-04-26

**Authors:** Shinpei Ikeda, Hiroshi Haga

**Affiliations:** 1Major of Occupational Therapy, Department of Rehabilitation, School of Health Sciences, Tokyo University of Technology, Tokyo 144-8535, Japan; 2Institute for Gerontology, J. F. Oberlin University, Tokyo 194-0294, Japan; 3School of Nursing, Saku University, Saku 385-0022, Japan

**Keywords:** COVID-19, older adults, depressive symptoms, stay-at-home, soft lockdown, lockdown, Japan

## Abstract

This study investigated the impact of stay-at-home orders on depressive symptoms among community-dwelling older adults during the early stages of the COVID-19 pandemic. A questionnaire was administered to older adults living in Ayase City, Kanagawa Prefecture, in July 2020, after the Japanese government declared its first COVID-19 state of emergency and stay-at-home order. In a sample of 1056 people, 69.1% were stay-at-home, and 30.9% were not. Those in the first group were more likely to be women, young-old, or non-workers. In addition, the patients tended to have more diseases. Of the participants, 39.3% had depressive symptoms and 60.7% did not. Multivariate analysis revealed that depressive symptoms were associated with increased frequency of being alone during the daytime (OR = 1.27; 95% CI = 1.07, 1.51), less face-to-face contact with friends or acquaintances (OR = 0.78; 95% CI = 0.65, 0.94), increased contact with friends or acquaintances through email/LINE app (OR = 1.29; 95% CI = 1.03, 1.60), and refraining from going out (OR = 1.54; 95% CI = 1.12, 2.09). These results suggest that quarantine measures related to soft lockdowns may aggravate the mental health of community-dwelling older adults. Therefore, it is necessary to consider macro-level policies.

## 1. Introduction

COVID-19 first broke out in Wuhan City, China, in December 2019 and has since spread worldwide. When Tedros Adhanom, the Director-General of the World Health Organization (WHO), declared COVID-19 a global pandemic on 11 March 2020, he called on countries worldwide to strengthen measures to prevent its spread [1]. In November 2022, more than two years after the declaration, Japan experienced the eighth wave of COVID-19, which showed no sign of convergence.

Back in April 2020, the government of Japan declared its first COVID-19 state of emergency and issued a “Stay-at-Home Order” to citizens in the greater Tokyo area. At the time, the legal force of behavioral restrictions in Japan was weaker than that in other countries. Consequently, the Japanese government’s response to the pandemic has been referred to as a soft lockdown [2].

In a healthcare framework, micro-, meso-, and macro-level factors influence an individual’s health [3]. Macro-level factors include social norms and values, policies, laws, regulations, and the overall structure of healthcare systems [3]. Therefore, quarantine measures against COVID-19, which call for individual behavioral restrictions, are macro-level factors that impact people’s lives and health. Thus, the relationship between the lives and health of the general population and government-initiated infectious disease policies should be examined.

In the early stages of the COVID-19 pandemic, there were concerns about the mental health impact of behavioral restrictions. In the United States (US), President Donald Trump declared a national emergency in March 2020 to prevent the spread of COVID-19 [4]. According to Liu and Mattke [5], people living in states with stay-at-home orders took preventive measures such as wearing masks and avoiding crowds. On the other hand, they experienced more mental distress, as assessed by the four-item version of the Patient Health Questionnaire (PHQ-4), than people who lived in states without stay-at-home orders [5]. A report focusing on mental health issues during the early stages of the COVID-19 pandemic indicated that quarantined people had a higher incidence of depression than the general population [6]. Based on the above findings, although behavioral restrictions could help reduce the spread of COVID-19, they can also have a negative effect on mental health.

In this study, we were deeply concerned about the mental health of community-dwelling older adults in Japan who are trying to cope with an aging population. As the majority of older Japanese adults refrained from going out in the early stages of the COVID-19 pandemic, they might have experienced more social isolation and feelings of loneliness than others at this time.

In Europe, which became the epicenter of COVID-19 in March 2020 [7], various governments tightened restrictions and introduced lockdowns to prevent the spread of COVID-19. According to several studies focusing on older adults, 62.7% practiced self-isolation in the United Kingdom (UK) when the lockdown was implemented on 23 March 2020 [8]. In a study surveying 993 older adults in Germany, 274 felt restricted under the lockdown, 411 did not feel restricted, and 308 felt neither restricted nor unrestricted [9]. In addition, survey results showed that feeling restricted was associated with depressive symptoms [9]. In the US, 79.3% of older adults responded that their social lives were restricted or negatively affected by stay-at-home orders; the study also demonstrated a relationship between loneliness and depression [10]. At the same time, in Mexico, where the government-ordered citizens to quarantine themselves, 57.6% of older adults only went out for basic needs such as shopping [11]. The report also found that older adults who complied with stay-at-home orders had higher incomes and more years of education than those who did not stay at home [11].

Other studies have found relationships between preventive behaviors against COVID-19 and age [12], sex [12,13], and health status [12] among older adults. However, no studies have focused on the Japanese government’s stay-at-home order and its relationship with the mental health of older Japanese adults.

It is well known that the general risk factors for depressive conditions among older adults are social factors such as having no one to talk to and not spending time with family; decreased social contact is associated with depression [14,15]. This suggests that restricting social interaction at the meso-level is related to the occurrence of depression in the context of the COVID-19 pandemic. Additionally, it is necessary to evaluate social systems and policies at the macro-level to understand how government-ordered self-quarantine affects older adults’ mental health.

Therefore, this study aimed to clarify the current status of older adults in Japan who refrained from going out during the early stages of COVID-19 and to investigate how this restraint was related to their depressive symptoms.

## 2. Materials and Methods

### 2.1. Study Design and Participants

In this study, we conducted a cross-sectional survey of community-dwelling older adults in Ayase City, Kanagawa Prefecture, located south of Tokyo, Japan. Kanagawa Prefecture is the second most populous prefecture in Japan after Tokyo, with approximately 9 million people residents. Ayase City is located on the plains of north-central Kanagawa Prefecture and has approximately 84,000 people, of which 23,000 are older adults in Ayase City.

The inclusion criteria were adults aged 65 years and older who did not require care in daily living, and participants (n = 2373) were selected by random sampling. Random sampling was conducted by municipal officers using the official residential registries that were maintained by the municipal administrations of Ayase City, including information such as age and sex. Data were collected through a survey sent by mail between July 18th and 10 August 2020, to investigate the effect of the early stages of COVID-19 (e.g., the first state of emergency from 7 April to 25 May 2020) on the lives and health of older adults in Japan. The questionnaire for this survey was prepared by the author and the municipal officers of Ayase City to obtain evidence for programs to maintain and improve older adults’ health after the COVID-19 pandemic (Appendix A). A total of 1343 people (response rate of 58.2%) answered the questionnaire, of whom 1114 people who did not have any deficiencies in the following variables, such as health status and social life, were included in the final analysis.

All procedures in this study were approved by the Research Review Board (Research Code: E20HS036) and the Ethics Committee of Tokyo University of Technology. The Mayor of Ayase City and the first author signed a Memorandum of Understanding (MOU) on the implementation of collaborative research and privacy protection of survey data.

### 2.2. Measurements

We assessed basic attributes such as sex, age, living arrangements, employment status, perceived economic status, and number of diseases. Age was divided into two groups: young-old (65–74 years) and old-old (aged 75 years and older). Living arrangements were divided into two groups: living alone and not living alone. Employment status was divided into two groups: workers and non-workers. Perceived economic status was categorized as poor, fair, or good. The number of diseases was characterized based on self-reported medical diagnoses, such as hypertension, stroke, diabetes, hyperlipidemia, Parkinson’s disease, depression, dementia, cancer, respiratory diseases, and musculoskeletal disorders.

To measure restraint from going out, we asked the question, “Do you refrain from going out?” If participants responded “yes,” they were then asked to justify their response by picking one of the following options: “illness,” “disability (after effects of stroke, etc.),” “pain in the legs,” “worry about using the toilet (incontinence, etc.),” “ear impairment (hearing problems, etc.),” “eye impairment,” “no fun outside,” “no transportation,” “prevention of new coronavirus infections,” or “other”.

To identify the presence or absence of depressive symptoms, we used a Two-question Screen: “Have you felt down or depressed in the past month?” and “Do you often feel that you have no interest in things or do not really enjoy anything?” If the participants responded “yes” to either of the two questions, they were classified as having depressive symptoms [16]. For older adults with major depression, the sensitivity and specificity are 96% and 57%, respectively, although the Two-question Screen is a short instrument for depression [17].

We investigated the participants’ social lives based on their frequency of being alone during the daytime and how often they had contact with friends or acquaintances. For the question “How often are you alone at home during the day?” responses were categorized as always, sometimes, or rarely. We measured the frequency of contact with friends or acquaintances in three situations (face-to-face, telephone, and email/LINE apps). Each situation was divided into three categories: every day to several times a week, several times a month, and several times a year to rarely.

### 2.3. Statistical Analysis

To measure restraint from going out, we first divided the participants into two groups: stay-at-home and non-stay-at-home groups. The stay-at-home group consisted of older adults who stayed at home to prevent the spread of COVID-19, while the non-stay-at-home group consisted of older adults who did not stay at home. Older adults who stayed at home for reasons other than to prevent COVID-19 were excluded from the study. Second, we evaluated the differences in the participants’ basic attributes compared with the second group to identify the characteristics of the stay-at-home group.

Third, we conducted analyses to assess the relationship between social life factors at the meso-level, refraining from going out at the macro-level, and depressive symptoms. In addition, continuous variables were expressed as mean ± standard deviation (SD) and compared using the Mann–Whitney U test. Categorical data are presented as percentages and were compared using the chi-square test. Finally, to determine whether social life factors and refraining from going out were associated with depressive symptoms, we used logistic regression analyses to predict each of these variables, using basic attributes as control variables. Statistical analyses were performed using IBM SPSS Statistics, version 25. The general level of significance was set at *p* < 0.05 (two tailed).

## 3. Results

### 3.1. Descriptive Analysis of Refraining from Going out

The mean age of the 1114 participants was 74.7 years (SD = 6.0; range, 65–101 years), with 47.6% being women. Of the participants, 788 (70.7%) said that they refrained from going out, whereas 326 (29.3%) did not. The most frequent reason for refraining from going out was to prevent COVID-19 infection (92.6%; n = 730), followed by pain in the legs (12.4%; n = 98) and illness (4.8%; n = 38).

In the final analysis, 730 people (69.1%) were classified into the stay-at-home group and 326 people (30.9%) into the non-stay-at-home group, totaling 1056 older adults in the final analysis.

Table 1 details the basic attributes of the stay-at-home and non-stay-at-home group. Those in the stay-at-home group were more likely to be women, young-old, and not working. In addition, they tended to have more diseases.

### 3.2. Relationship between Depressive Symptoms, Social Life, and Refraining from Going out

Of the 1056 older adults, 39.3% had depressive symptoms and 60.7% did not. We examined the relationships between depressive symptoms, social life factors, and refraining from going out (Table 2). Between the two groups, there were significant differences for the following variables: frequency of being alone during the daytime, frequency of face-to-face contact with friends or acquaintances, and refraining from going out. Participants who were always alone during the day had more depressive symptoms than those who reported having these feelings occasionally or rarely (35.4% vs. 30.4% and 34.2%, respectively). Participants who had face-to-face contact with friends or acquaintances several times a year or rarely had more depressive symptoms than those who reported having contact every day, several times a week, or several times a month (38.3% vs. 32.8% and 28.9%, respectively). Participants who refused to go out had more depressive symptoms than those who did not (77.1% vs. 64.0%).

As shown in Table 3, logistic regression analyses were used to explain the influence of social life factors and refraining from going out on depressive symptoms among older adults. The results showed that a high frequency of being alone in the daytime (OR = 1.27; 95% CI = 1.07, 1.51), low frequency of face-to-face contact with friends or acquaintances (OR = 0.78; 95% CI = 0.65, 0.94), high frequency of contact with friends or acquaintances via the email/LINE app (OR = 1.29; 95% CI = 1.03, 1.60), and refraining from going out (OR = 1.54; 95% CI = 1.12, 2.09) were associated with more depressive symptoms.

## 4. Discussion

### 4.1. Refraining from Going out and the Prevalence of Depressive Symptoms

Japan’s government experienced its first COVID-19 state of emergency between 7 April and 25 May 2020; throughout June, the government did not take any specific measures to regulate people’s behaviors. The survey was conducted from 18 July to 10 August 2020, and 69.1% of older adults continued to stay at home during this period.

In the UK, previous research found that 62.7% of older adults self-isolated during the lockdown period [8], whereas in Mexico, this figure was 57.6% [11]. This study found that 70% of the older adults in Japan refrained from going out during the study period. This seems to have been affected by the government’s call for a new lifestyle during the COVID-19 pandemic, including social distancing measures after the first state of emergency.

Women, characterized as young or old, who did not work, and those with more diseases were likely to be in the stay-at-home group (Table 1). Regarding the relationship between the number of diseases and refraining from going out, elderly people with multiple diseases refrained from going out to avoid severe infections. Previous studies have pointed out that COVID-19 infection prevention behaviors are more likely to be practiced by women than men [12,13], which is in accordance with the results presented here. During the study period, young-old people were more likely to refrain from going out than were old-old people. Because young people have many opportunities to play social roles, it seems that the young-old responded to requests to refrain from going out even after the first state of emergency ended. Compared with working participants, non-workers had fewer opportunities to go out, making it easier for them to refrain from going out.

The results of this study showed that 39.3% of participants had depressive symptoms. Other studies found that 44% of elderly people in Spain, France, and Italy were in a state of depression during the lockdown [18] in March 2020. Regardless of the legal force of behavioral restrictions, stay-at-home orders may be a threat that makes it easy for older adults to experience depressive symptoms.

### 4.2. Association between Behavioral Restrictions and Depressive Symptoms

We conducted a univariate analysis to examine the relationship between depressive symptoms, social life factors, and refraining from going out. The frequency of being alone during the daytime, having face-to-face contact with friends or acquaintances, and refraining from going out were associated with depressive symptoms. Furthermore, logistic regression analysis found that the factors that independently affected depressive symptoms were an increased frequency of being alone in the daytime, decreased frequency of face-to-face contact with friends or acquaintances, and an increased frequency of contact with friends or acquaintances via the email/LINE app and refraining from going out.

In older adults, spending little time with family members and having no opportunity to talk to others are general risk factors for depression [14,15]. Consequently, declining social interaction has become a serious problem during the COVID-19 pandemic. In addition, it should be stressed that prevention measures such as stay-at-home orders in the early stages of the COVID-19 pandemic could have promoted depression among older adults. In Germany, feeling restricted by the COVID-19 lockdown have been associated with depressive symptoms among older adults [9]. Even the participants in our study, we assumed, experienced the sense of restriction related to the soft lockdown such as refraining from going out, to their depressive symptoms.

According to Sepúlveda-Loyola et al., interacting with friends and relatives using technology helped older adults maintain their mental health during the COVID-19 pandemic [18]. However, the multivariate analysis in our study revealed that email and/or LINE promoted depressive symptoms. Therefore, it is necessary to further investigate how social interactions that do not occur face-to-face affect the mental health of older adults.

On 25 May 2020, when Japan’s government lifted its first COVID-19 state of emergency, Prime Minister Abe announced to the national and international mass media that the government of Japan had succeeded in bringing the early stages of the COVID-19 pandemic almost to an end earlier than other countries. In addition, he evaluated the soft lockdown, known as the “Japan Model,” very highly [19]. In Japan, however, it has been pointed out that the nationwide school closure request on 27 February 2020 had a significant impact on the public in terms of fostering a sense of crisis regarding COVID-19. In other words, restrictions on socioeconomic activities were tightened before the first state of emergency, and quarantine measures were introduced in Japan earlier than in Europe or the US. Therefore, adopting a non-mandatory soft lockdown for approximately three months at the macro-level had a negative impact on the mental health of older adults in Japan. Governments should consider this finding when discussing the future implementation of quarantine measures.

### 4.3. Strengths and Limitations

In the context of the COVID-19 pandemic, this study provides initial evidence of the effects of soft lockdowns on depressive symptoms in community-dwelling older adults. However, this study had several limitations. First, we conducted an analysis using a cross-sectional survey. Although the participants were selected by random sampling, they lived in an area near Tokyo. Future research should collect data from other areas in greater Tokyo to gain a wider view of older adults’ experiences and analyze how this effect changes over time. Second, we analyzed social life and restraint data from surveys conducted by municipal administrations in Ayase City. Consequently, we were unable to consider other relevant factors that may be associated with depressive symptoms in community-dwelling older adults during the COVID-19 soft lockdown, such as COVID-19-related stress and coping [20], life events related to COVID-19, or knowledge and opinions about COVID-19. Third, an advantage of using a screening instrument for depression, such as the Two-question Screen, is that the questions are simple and self-reported by participants; however, screening tools cannot be used in comprehensive clinical interviews to confirm the diagnosis of depression. Moreover, several countries that implemented lockdowns also provided COVID-19-related financial aid to businesses and households. Therefore, it is necessary to examine the relationship between economic assistance, citizens’ restraint from going out, and the prevalence of depressive symptoms.

## 5. Conclusions

Based on an analysis of 1056 people, our study revealed that 69.1% and 30.9% of older adults belonged to the stay-at-home and non-stay-at-home groups, respectively. Of the total sample, 39.3% had depressive symptoms, and 60.7% did not. Depressive symptoms were associated with an increased frequency of being alone during the daytime, decreased frequency of face-to-face contact with friends or acquaintances, increased frequency of contact with friends or acquaintances via the email/LINE app, and refraining from going out. During the early stages of the COVID-19 pandemic, quarantine measures required people to stay at home as part of a soft lockdown; restraint from going out could have aggravated the mental health of older adults.

## Figures and Tables

**Table 1 healthcare-11-01239-t001:** Basic attributes and refraining from going out.

		Total n (%)	Refraining from Going out		*p*-Value
			Stay-at-Home n (%)	Non-Stay-at-Home n (%)	
sex	man	549 (52.0)	330 (45.2)	219 (67.2)	<0.001
	woman	507 (48.0)	400 (54.8)	107 (32.8)	
age	young-old	571 (54.1)	412 (56.4)	159 (48.8)	0.021
	old-old	485 (45.9)	318 (43.6)	167 (51.2)	
living arrangement	living alone	144 (13.6)	91 (12.5)	53 (16.3)	0.097
	not living alone	912 (86.4)	639 (87.5)	273 (83.7)	
employment status	workers	308 (29.2)	182 (24.9)	126 (38.7)	<0.001
	non-workers	748 (70.8)	548 (75.1)	200 (61.3)	
perceived economic status	good	160 (15.2)	109 (14.9)	51 (15.6)	0.956
	fair	697 (66.0)	483 (66.2)	214 (65.6)	
	poor	199 (18.8)	138 (18.9)	61 (18.7)	
number of diseases	mean ± SD	1.5 ± 1.3	1.6 ± 1.3	1.3 ± 1.2	0.001

**Table 2 healthcare-11-01239-t002:** Comparison of social life factors and refraining from going out between two groups of depressive symptoms.

		Total n (%)	Depressive Symptoms		*p*-Value
			Yes n (%)	No n (%)	
frequency of being alone in the daytime	always	306 (29.0)	147 (35.4)	159 (24.8)	< 0.001
	sometimes	314 (29.7)	126 (30.4)	188 (29.3)	
	rarely	436 (41.3)	142 (34.2)	294 (45.9)	
frequency of contact with friends or acquaintances					
face-to-face	every day to several times a week	390 (36.9)	136 (32.8)	254 (39.6)	0.023
	several times a month	310 (29.4)	120 (28.9)	190 (29.6)	
	several times a year to rarely	356 (33.7)	159 (38.3)	197 (30.7)	
by telephone	every day to several times a week	392 (37.1)	154 (37.1)	238 (37.1)	1.000
	several times a month	346 (32.8)	136 (32.8)	210 (32.8)	
	several times a year to rarely	318 (30.1)	125 (30.1)	193 (30.1)	
by email/LINE app	every day to several times a week	358 (33.9)	153 (36.9)	205 (32.0)	0.187
	several times a month	274 (25.9)	108 (26.0)	166 (25.9)	
	several times a year to rarely	424 (40.2)	154 (37.1)	270 (42.1)	
refraining from going out	stay-at-home	730 (69.1)	320 (77.1)	410 (64.0)	<0.001
	non-stay-at-home	326 (30.9)	95 (22.9)	231 (36.0)	

**Table 3 healthcare-11-01239-t003:** Results of the logistic regression analysis: associations between social life factors and refraining from going out with depressive symptoms.

	Depressive Symptoms a		
	OR	CI	*p*-Value
frequency of being alone in the daytime b	1.27	1.07–1.51	0.006
frequency of contact with friends or acquaintances			
face-to-face c	0.78	0.65–0.94	0.010
by telephone c	0.93	0.73–1.18	0.560
by email/LINE app c	1.29	1.03–1.60	0.026
refraining from going out d	1.54	1.14–2.09	0.005

Odds ratio was adjusted for gender, age, living arrangement, employment status, perceived economic status, and number of diseases. a. the responses are scored as: 2 = yes and 1 = no. b. the responses are scored as: 3 = always, 2 = sometimes, and 1 = rarely. c. the responses are scored as: 3 = every day to several times a week, 2 = several times a month, and 1 = several times a year to rarely. d. the responses are scored as: 2 = stay-at-home and 1 = non-stay-at-home.

## Data Availability

The datasets produced and analyzed in this study are not publicly available.

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
