# Peer review of "The Impact of Japan’s Soft Lockdown on Depressive Symptoms among Community-Dwelling Older Adults"

_healthcare, 2023, doi:10.3390/healthcare11091239_

Round 1

Reviewer 1 Report

The topic of the research is interesting and relevant: „ This study investigated the impact of Japan’s stay-at-home order on depressive symptoms 10 among community-dwelling older adults during the early stages of the COVID-19 pandemic.” (Lines 10-11)

Its purpose is precisely defined: " the purpose of this study was to clarify the current status of older adults in Japan who refrained from going out during the early stages of COVID-19 and to investigate how this restraint relates to their depressive symptoms " (Lines 88-90) The presentation of the previous works related to the research area is a bit short, this part should definitely be expanded (line 34-87).

I would like to recommend the Social Stress Process Model (Pearlin, Leonard I., et al. "The stress process." Journal of Health and Social behavior (1981): 337-356.; McLeod, Jane D. "The meanings of stress: Expanding the stress process model." Society and Mental Health 2.3 (2012): 172-186.). Several studies have used the Social Stress Process Model during research related to the Covid-19 epidemic (Nelson, Niccole A., and Cindy S. Bergeman. "Daily stress processes in a pandemic: The effects of worry, age, and affect." The Gerontologist 61.2 (2021): 196-204.).

The research methodology is a questionnaire survey, which is scientifically appropriate. The researchers 1343 responses were received and analyzed. The execution of the analysis and the presentation of the results are scientifically appropriate. The research results are remarkable. It is particularly important that ethical regulations were followed during the research. (Lines 104-105)

Overall, I recommend publishing the study with minor corrections (expansion of relevant research).

Author Response

添付ファイルをご覧ください。

Reviewer 2 Report

The impact of quarantines on the mental health of those populations affected is worthy of consideration. The extent to which older generations are more susceptible to mental health issues than younger generations is unclear in the present study. Perhaps a follow up study on such issues in the younger generations would shed some light on the nature of mental health risk during a quarantine.

A major concern of mine with regard to this paper is that you seem to be employing a past survey in order to assess present mental health. You write, “In this study, we were deeply concerned about the mental health of community-dwelling older adults in Japan, which is trying to cope with the aging of its population. As the majority of Japanese older adults refrained from going out in the early stages of the COVID-19 pandemic, they may now experience more social isolation and feelings of loneliness than before.” Elsewhere you write, “the purpose of this study was to clarify the current status of older adults in Japan who refrained from going out during the early stages of COVID-19 and to investigate how this restraint relates to their depressive symptoms.” I don’t believe the paper addressed this issue. Your data speaks to the mental health of older adults in 2020, not 2022 or 2023. Are you suggesting that measuring mental health in 2020 has lingering significance for mental health today? You may want to clarify this. A similar note of clarification is warranted for “community-dwelling” adults… what does it mean to be community-dwelling?

Though adequately described, I believe your method for identifying depressive symptomatology could be a bit more rigorous. Can we really say that someone who felt down, perhaps only once, in the past month establishes a profile of depression? It would seem that a more extensive survey could help make the case for which you argue. The other contributing factors to depression identified here all pertain to social isolation; this is nothing terribly new. Surely, quarantine could contribute to social isolation and the attending depression. In fact, that is your tentative conclusion: “During the early stages of the COVID-19 pandemic, quarantine measures requiring people to stay at home as part of a soft lockdown could have aggravated the mental health of older adults.” Of course, it could have, but did it? Such an uncertain conclusion is not terribly compelling. I believe you need to clarify further your conclusion. You write, “Based on the analysis of 1056 people, our study revealed that 69.1% and 30.9% of older adults were part of the stay-at-home and non-stay-at-home groups, respectively. A total of 39.3% had depressive symptoms and 60.7% did not.” 39.3% of the total sample? Or, 39.3% of those who stayed home?

Lastly, I found the following paragraph unclear: “In Japan, it has been pointed out that the nationwide school closure request on February 27, 2020, had a significant impact on the public in terms of fostering a sense of crisis regarding COVID-19. In other words, restrictions on socioeconomic activities were tightened before the 1st state of emergency, and quarantine measures were introduced in Japan earlier than in Europe and the United States.” What does school closure have to do with socioeconomic activities? Why mention Japan’s quarantine measures relative to those in Europe and the United States? Why is that significant?

Overall, this paper addresses some interesting issues but it is not terribly original or insightful.

Reviewer 3 Report

Thank you for inviting me as a reviewer of this valuable manuscript. I recommend following suggestions for improving the quality of manuscript.

(Comment 1) The authors simply presented the results of a survey of older adults. It is difficult to say that the analysis results of this study have identified the relationship between stay-at-home and depressive symptoms. Therefore, I recommend authors to conduct additional analysis based on the survey data to derive statistical significance as to why stay-at-home causes depressive symptoms compared to non-stay-at-home.

Method

(Comment 2) I recommend authors to supplement basic characthteristics of Kanagawa Prefecture.

(Comment 3) Random sampling selection process is unlcear. Are these samples representative of ‘Kanagawa Prefecture’? Is the randomly selected sample data controlled by the conutry level? This is an important point to be able to generalize the findings of study.

(Comment 4) The author used personal information (e.g. e-mail, age) for the survey. Are you obtained all privacy agreements in advance?

(Comment 5) I recommend authors to supplement 'Questionnaire development process'. In addition, the final questionnaire must be submitted as an attachment.

Reviewer 4 Report

This paper is to examine the impact of the lockdown policy on mental health among community-dwelling older adults during COVID-19. This Article can increase our knowledge of the extent to which lockdown affects the health of older people. But before publishing, there are still some questions that need to be clarified.

1. the authors only use two questions to identify depressive symptoms. Is it scientific accuracy?

2. in the present version, readers can only find some social activity data. But readers can not find the data on the participant’s knowledge and opinion about COVID-19 and lockdown policy. The reader can’t if some participants have experienced some events related to COVID-19 such as illness and loss of family members and close friends.

Round 2

Reviewer 4 Report

The author has responded well to my questions, so I propose to publish the article.